# SRBGCN: Tangent space-Free Lorentz Transformations for Graph Feature Learning

## Abstract

Hyperbolic graph convolutional networks have been successfully applied to represent complex graph data structures. However, optimization on Riemannian manifolds is nontrivial thus most of the existing hyperbolic networks build the network operations on the tangent space of the manifold, which is a Euclidean local approximation. This distorts the learnt features, limits the representation capacity of the network and makes it hard to optimize the network. In this work, we introduce a fully hyperbolic graph convolutional network (**GCN**), referred to as **SRBGCN**, which performs neural computations such as feature transformation and aggregation directly on the manifold, using manifold-preserving Lorentz transformations that include spatial rotation (**SR**) and boost (**B**) operations. Experiments conducted on static graph datasets for node classification and link prediction tasks validate the performance of the proposed method.

## 1 Introduction

Graph convolutional networks (GCNs) were proposed to make use of the graph topology and model the spatial relationship between graph nodes, hence generalizing the convolution operation to graph data[Kipf & Welling (2017); Defferrard et al. (2016)]. Initially, the proposed models were built in the Euclidean space Hamilton et al. (2017); Zhang et al. (2018); Velickovic et al. (2019) which is not the natural space for embedding graph data and produces distorted feature representations Nickel & Kiela (2018); Chami et al. (2019). Hyperbolic spaces are more suitable for representing graph data as the space volume is increasing exponentially which is perfect for embedding tree-like data structures that also grow exponentially with the depth of the tree whereas the space grows polynomially for the Euclidean space. Motivated by this, recent works built GCNs in the hyperbolic space to take advantage of the hyperbolic geometry properties Chami et al. (2019); Liu et al. (2019). The hyperbolic graph convolutional networks (HGCNs) achieved better performance than the corresponding Euclidean ones which shows the effectiveness of using the hyperbolic space to model hierarchical data structures and graph data.

However, these works performed the network operations in the tangent space of the manifold which is a Euclidean local approximation to the manifold at a point. The Euclidean network operations such as feature transformation and feature aggregation are not manifold-preserving and can not be directly applied on the manifold, that is why these methods resort to the tangent space. However, using a tangent space may limit the representation capabilities of the hyperbolic networks which is caused by distortion specially as most of these works used the tangent space at the origin. In this work, we propose a full manifold-preserving Lorentz feature transformations using both boost and spatial rotation operations to build SRBGCN fully in the hyperbolic space without resorting to the tangent space. Experiments conducted on node classification and link prediction tasks on static graph datasets show the effectiveness of our proposed method. SRBGCN has a good physical interpretation and can be used to build deep networks with more representation capacity and less distorted features.

## 2 Related Work

Chami et al. (2019) proposed HGCNs where networks operations are performed in the tangent space of the manifold. They were able to achieve better performance than the Euclidean analogs on node

classification and link prediction tasks. Concurrently to this work, Liu et al. (2019) proposed the Hyperbolic Graph Neural Networks (HGNNs) which performed well on graph classification tasks. Several models have been proposed using different hyperbolic models especially the Lorentz and the Poincaré models for different other tasks such as image segmentation Gulcehre et al. (2018), word embeddings Tifrea et al. (2018), human action recognition Peng et al. (2020), text classification Zhu et al. (2020), machine translation Shimizu et al. (2020); Gulcehre et al. (2018), knowledge graph embeddings Chami et al. (2020) and so on. Gu et al. (2018) built a two-stream network for Euclidean and hyperbolic features (operations on tangent space) and used an interaction module to enhance the learnt feature representations in the two geometries. Peng et al. (2021) presented a comprehensive survey for hyperbolic networks.

Zhang et al. (2021b) rebuilt the network operations in HGCNs to guarantee that the learnt features follow the hyperbolic geometry and used the Lorentz centroid Ratcliffe et al. (1994); Law et al. (2019) for aggregating the features. Zhang et al. (2021a) used attention modules in the hyperbolic space to build hyperbolic networks. Dai et al. (2021) built a hyperbolic network by imposing the orthogonal constraint on a sub-matrix of the transformation matrix (subspace transformation). They used the same number of learnable parameters for the feature transformation step as the networks built on the tangent space, however the orthogonal constraint ensured that the transformation is manifold-preserving and they did not need to learn the parameters on the tangent space. They used the Einstein midpoint method defined in the Klein model Ungar (2005) for the feature aggregation step. Chen et al. (2022) used a normalization procedure to keep the points on the manifold. They also use normalization for the feature aggregation step. The idea is similar to learning a general transformation matrix or performing aggregation for spherical embeddings and then normalizing the resulting features to have the norm of the sphere radius. In this work, we introduce a full space manifold-preserving transformation matrix in SRBGCN without the need for normalization to keep the points on the manifold.

## 3 BACKGROUND

### 3.1 GRAPH CONVOLUTIONAL NETWORKS

A static graph $\mathcal{G} = \{\mathcal{V}, \mathcal{E}\}$ where $\mathcal{V} = \{v_1, v_2, \ldots, v_n\}$ is the set of $n$ graph nodes with $\mathcal{E}$ representing the set of graph edges. The edge set $\mathcal{E}$ can be encoded in an adjacency matrix $\mathbf{A} \in \mathbb{R}^{n \times n}$ where $\mathbf{A}_{i,j} \in [0,1]$ if there is a link between $v_i$ and $v_j$ otherwise, $\mathbf{A}_{i,j} = 0$. Each node $v_i$ has a feature vector in the Euclidean space $x_i \in \mathbb{R}^d$ of dimension $d$ and $\mathbf{X}$ is the set of features for all the n nodes in the graph.

The feature transformation step in GCNs can be formulated as:

$$\mathbf{Y}^l = \mathbf{X}^l \mathbf{W}^l + \mathbf{B}^l \tag{1}$$

where $\mathbf{W}^l$ is the weight matrix corresponding to the input $\mathbf{X}^l$ at layer $l$ and $\mathbf{B}^l$ is the bias translation matrix. The weight matrix acts as a linear transformer whereas the optional bias matrix makes the transformation affine.

Then the feature aggregation from neighboring nodes step with nonlinear activation applied can be formulated as:

$$\mathbf{X}^{l+1} = \sigma(\mathbf{D}^{-1/2}(\mathbf{A} + \mathbf{I})\mathbf{D}^{-1/2}\mathbf{Y}^l) \tag{2}$$

where $\sigma$ is an activation function. $\mathbf{D}^{-1/2}(\mathbf{A} + \mathbf{I})\mathbf{D}^{-1/2}$ is the normalized adjacency matrix to normalize nodes weights in the neighboring set. $\mathbf{D}$ is a diagonal matrix where $\mathbf{D}^{ii} = 1 + \sum_j A^{ij}$ and $\mathbf{I}$ is the identity matrix to keep identity features. $\mathbf{X}^{l+1}$ represents the output of layer $l$ which can be the input to the next layer $l + 1$. A GCN is built by stacking a number of those layers.

Clearly, the linear transformation matrix $W$ can not be used in hyperbolic networks as this unconstrained transformation matrix will not keep points on the manifold i.e. not manifold preserving transformations. The same applies for the aggregation step as the Euclidean mean operation is not manifold-preserving.

Figure 1: Regular/ circular rotation vs hyperbolic rotation/ squeeze mapping. The axes and the points are color coded for illustration purposes.

## 3.2 HYPERBOLIC ROTATIONS/SQUEEZE MAPPING

A regular rotation is a linear map that preserves the Euclidean inner product $\langle .,. \rangle_{\mathcal{E}} : \mathbb{R}^d \times \mathbb{R}^d \to \mathbb{R}$ where $\langle x, y \rangle_{\mathcal{E}} := \sum_{i=0}^{d-1} x_i y_i$ whereas a hyperbolic rotation or a squeeze mapping is a linear map that preserves $\langle .,. \rangle_{\mathcal{L}}$. Regular rotations can be realized by trigonometric functions whereas hyperbolic rotations can be realized by hyperbolic functions which are related to their trigonometric counterparts through complex angles. Intuitively, regular rotations can be thought of as a regular rotation to the axes whereas hyperbolic rotations are rotations in the hyperbolic sense and can be thought of as squeezing the axes (see Figure 1 for visualization), hence the name. The following matrix is a squeeze mapping that keeps the points rotating on a hyperbola ($\mathbb{H}^{1,K}$):

$$\mathbf{L}(\omega) = \begin{bmatrix} \cosh \omega & \sinh \omega \\ \sinh \omega & \cosh \omega \end{bmatrix} \tag{3}$$

A hyperbolic geometry review is provided in the appendix to make the paper self-contained.

## 4 SPATIAL ROTATION BOOST GRAPH CONVOLUTIONAL NETWORK(SRBGCN)

In this part, we show how to build SRBGCN fully in the hyperbolic space. A manifold-preserving Lorentz transformation is used for feature transformation that includes the boost and spatial rotation operations.

### 4.1 LORENTZ TRANSFORMATION

The linear transformation matrix used in Equation 1 for Euclidean features can not be used directly for hyperbolic features as this linear transformation unconstrained matrix is not manifold-preserving for the hyperbolic space. Instead, a Lorentz transformation matrix $\Lambda$ should satisfy the following constraint:

$$\Lambda^T g_{\mathcal{L}} \Lambda = g_{\mathcal{L}} \tag{4}$$

where $\Lambda \in \mathbb{R}^{(d+1) \times (d+1)}$ and $T$ represents the transpose operation of the matrix. A Lorentz transformation matrix is a matrix that is orthogonal with respect to the Minkowski metric $g_{\mathcal{L}}$ and belongs to the Lorentz group. When $\Lambda_0^0$ is positive (the first element in the transformation matrix), the mapping remains on the upper half of the hyperboloid. Taking the determinant of this equation, we obtain $(\det \Lambda)^2 = 1$ i.e. $(\det \Lambda) = \pm 1$. The set of matrices $\Lambda$ with $(\det \Lambda) = 1$ and $\Lambda_0^0 > 0$ are referred to as the proper Lorentz group $SO(d,1)^+$.

### 4.2 BOOST AND SPATIAL ROTATION

The Lorentz transformation can be decomposed into a boost and a spatial rotation operations Durney (2011); Moretti (2002) by polar decomposition. The boost matrix is symmetric semi-positive and the spatial rotation matrix is unitary. The spatial rotation operation rotates the spatial coordinates and the boost operation moves a point along the time coordinate without rotating the spatial coordinates.

Intuitively, the subspace manifold formed by the spatial axes is a $(d-1)$-dimensional sphere for a $d$-dimensional hyperboloid at a given level $\hat{x_0} \in x_0 > K$ since $\sum_{i=1}^{d} x_i^2 = \hat{x_0}^2 - K$ is a sphere for any value $\hat{x_0} \in x_0 > K$. Hence, a regular rotation transformation matrix represents the spatial rotation operation in this subspace manifold. The spatial rotation matrix is given by:

$$\mathbf{P} = \begin{bmatrix} 1 & 0 \\ 0 & \mathbf{Q} \end{bmatrix}_{(d+1)\times(d+1)} \tag{5}$$

where $\mathbf{Q}$ belongs to the special orthogonal group $SO(d)$ i.e. $\mathbf{Q^T Q = I}$. It can easily verified that $\mathbf{P}$ is a Lorentz transformation matrix which satisfies Equation 4.

**Proof:** From Equation 4, we have:

$$\mathbf{P}^T g_\mathcal{L} \mathbf{P} = \begin{bmatrix} 1 & 0 \\ 0 & \mathbf{Q}^T \end{bmatrix} \begin{bmatrix} -1 & 0 \\ 0 & \mathbf{I} \end{bmatrix} \begin{bmatrix} 1 & 0 \\ 0 & \mathbf{Q} \end{bmatrix} = \begin{bmatrix} -1 & 0 \\ 0 & \mathbf{Q}^T \mathbf{I} \mathbf{Q} \end{bmatrix} = \begin{bmatrix} -1 & 0 \\ 0 & \mathbf{I} \end{bmatrix} = g_\mathcal{L}$$

The rotation matrix $\mathbf{Q}$ in Equation 5 can be realized using different representations such as the basic rotations using Trigonometric functions ($d$ degrees of freedom in this case), axis and angle ($d + 1$ degrees of freedom) or using the Gram-Schmidt orthonormalization process. We enforce the orthogonalization constraint on the spatial rotation feature transformation matrix as the angles in the other two methods form a discontinuous search space Zhou et al. (2019) and also there are singularities in this search space (gimbal lock problem).

The boost operation can be realized using hyperbolic rotation or squeeze mapping. Since the squeeze mapping matrix in Equation 3 satisfies the constraint 4, we can use such transformation matrix in the hyperbolic feature transformation step. A $d$ basic hyperbolic rotations as in Equation 3 for each spatial axis with the time axis can be used to realize the boost operation. A more compact hyperbolic rotation representation using a hyperbolic rotation axis $n_d$ and a hyperbolic rotation parameter $\omega$ (regular circular rotation can be realized also using an axis and an angle for rotation) in a $d$-dimensional hyperboloid can be represented by the transformation matrix:

$$\mathbf{L} = \begin{bmatrix} \cosh\omega & (\sinh\omega)n_d^T \\ (\sinh\omega)n_d & \mathbf{I} - (1 - \cosh\omega)n_d \otimes n_d \end{bmatrix}_{(d+1)\times(d+1)} \tag{6}$$

where $\otimes$ represents the outer product operation. The hyperbolic rotation plane is the plane parallel to the plane spanned by the normal vector $n_d$ (a normalized linear combination of the spatial axes) and the axis $x_0$ (referred to as the time axis in special relativity). Note that when $n$ is a canonical basis vector, the resulting matrix will be similar to the one in Equation 3 after padding with zeros for other canonical spatial basis vectors.

**Proof:** Let $\mathbf{L} = \begin{bmatrix} a & b_d^T \\ b_d & \mathbf{C}_{d\times d} \end{bmatrix}$ where $\mathbf{C} = \mathbf{C}^T$ as $\mathbf{L}$ is a symmetric matrix.

From Equation 4, we have:

$$\mathbf{L}^T g_\mathcal{L} \mathbf{L} = \begin{bmatrix} a & b_d^T \\ b_d & \mathbf{C}^T \end{bmatrix} \begin{bmatrix} -1 & 0 \\ 0 & \mathbf{I} \end{bmatrix} \begin{bmatrix} a & b_d^T \\ b_d & \mathbf{C} \end{bmatrix} = \begin{bmatrix} a & b_d^T \\ b_d & \mathbf{C} \end{bmatrix} \begin{bmatrix} -a & -b_d^T \\ b_d & \mathbf{C} \end{bmatrix}$$

$$= \begin{bmatrix} -a^2 + b_d \cdot b_d & -ab_d + \mathbf{C}b_d \\ -ab_d + \mathbf{C}b_d & -b_d \otimes b_d + \mathbf{C}^2 \end{bmatrix} = g_\mathcal{L} = \begin{bmatrix} -1 & 0 \\ 0 & \mathbf{I} \end{bmatrix}$$

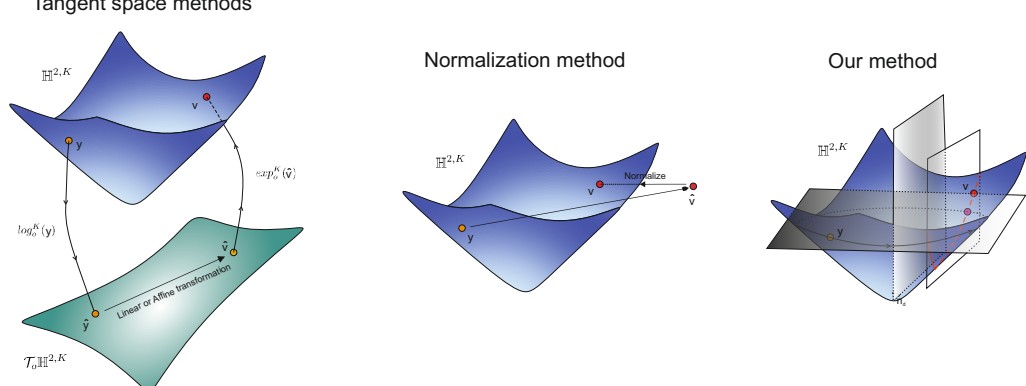

Figure 2: Left: the transformation step in the methods that rely on the tangent space of the manifold. Middle: the normalization method used for feature transformation in Chen et al. (2022). Right: our transformation method using a circular regular rotation for the spatial rotation operation and a hyperbolic rotation for the boost operation.

So, we get $-a^2 + b_d \cdot b_d = -1$ and using the hyperbolic identity: $-\cosh^2 \omega + \sinh^2 \omega = -1$, we have $a = \cosh \omega$ and $b_d = (\sinh \omega)n_d$ where $n_d$ is a unit vector. To solve for $\mathbf{C}$, we have $-b_d \otimes b_d + \mathbf{C}^2 = \mathbf{I}$. So, we get:

$$\mathbf{C}^2 = \mathbf{I} + (\sinh^2 \omega)n_d \otimes n_d = \mathbf{I} + (-1 + \cosh^2 \omega)n_d \otimes n_d$$

$$= \mathbf{I} + (1 - 2 + \cosh^2 \omega + 2\cosh \omega - 2\cosh \omega)n_d \otimes n_d$$

$$= \mathbf{I} + (1 - \cosh \omega)^2 n_d \otimes n_d - 2(1 - \cosh \omega)n_d \otimes n_d = (\mathbf{I} - (1 - \cosh \omega)n_d \otimes n_d)^2$$

and using $-ab_d + \mathbf{C}b_d = 0$, we omit the negative solution and accept the positive one. So, we have $\mathbf{C} = \mathbf{I} - (1 - \cosh \omega)n_d \otimes n_d$ as the solution.

Since $\mathbf{L}$ and $\mathbf{P}$ represent the boost and spatial rotation operations, respectively. We use the following Lorentz transformation matrix as the feature transformation matrix for the hyperbolic space:

$$\mathbf{M} = \mathbf{PL} \tag{7}$$

To show that $\mathbf{M}$ is a Lorentz transformation matrix, we have $(\mathbf{PL})^T g_{\mathcal{L}}(\mathbf{PL}) = \mathbf{L}^T(\mathbf{P}^T g_{\mathcal{L}}\mathbf{P})\mathbf{L} = \mathbf{L}^T g_{\mathcal{L}}\mathbf{L} = g_{\mathcal{L}}$ since both $\mathbf{P}$ and $\mathbf{L}$ are Lorentz transformation matrices as shown before.

Figure 2 shows a visualization comparison between different methods. SRBGCN is fully hyperbolic as the boost and spatial rotation operations are intrinsic manifold-preserving transformation.

### 4.3 FEATURE TRANSFORMATION AND AGGREGATION IN SRBGCN

The feature transformation step in SRBGCN in contrast to the Euclidean one in Equation 1 can be formulated as:

$$\mathbf{Y}_h^l = \mathbf{X}_h^l \mathbf{M}^l = \mathbf{X}_h^l \mathbf{P}^l \mathbf{L}^l \tag{8}$$

where the subscript $h$ represents features on the hyperboloid. To get the initial features representation on the hyperboloid, the exponential map (Equation 12) at the origin is used to map the features from the tangent space at the origin to the hyperboloid.

For the feature aggregation step, we use the Lorentz centroid Ratcliffe et al. (1994); Law et al. (2019) which minimizes the squared Lorentzian distance. It can be computed as:

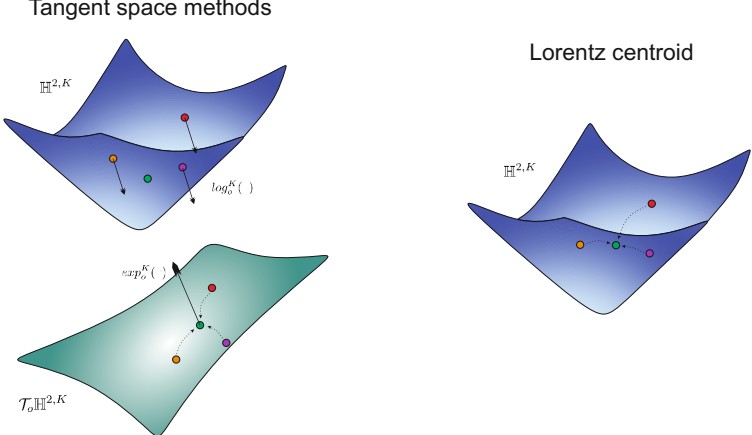

Figure 3: Left: the aggregation step in the tangent space methods. Right: the Lorentz centroid.

$$x_i^{h,l+1} = \sqrt{K} \frac{\sum_{j \in NS(i)} w_{i,j} y_j^{h,l}}{|\|\sum_{j \in NS(i)} w_{i,j} y_j^{h,l}\|_{\mathcal{L}}|} \tag{9}$$

where $-1/K$ is the constant negative curvature ($K > 0$), $NS(i)$ is the neighbor set for node $i$ which includes the node itself and $w_{i,j}$ is the weight between nodes $i$ and $j$ in the normalized adjacency matrix. Figure 3 shows a visualization of different aggregation techniques used in hyperbolic networks.

The margin ranking loss is used as the loss function defined as:

$$Loss = max(d - \hat{d} + m, 0) \tag{10}$$

where $m$ is the non-negative margin hyperparameter. For node classification tasks where the goal is to predict the label of a given node in a graph, $d$ is the distance between a node and its correct class whereas $\hat{d}$ is distance between the node and wrong class. We calculate the hyperbolic distances between the node and a set of trainable centroids on the manifold and feed the distances vector to a softmax classifier for node classification. For link prediction tasks where the goal is to predict the existence of links between graph nodes, $d$ is the distance between connected nodes where links exist and $\hat{d}$ is the distance for negative samples. We use the Fermi-Dirac decoder Tifrea et al. (2019) to calculate the probability for link existence between nodes.

## 5 EXPERIMENTS

Experiments are conducted on four publicly available datasets: Disease, Airport, PubMed and Cora. The datasets description, evaluation metrics and the evaluation results with comparison to other methods are presented in the following subsections.

### 5.1 DATASETS DESCRIPTION

The Disease dataset is constructed by using the SIR disease spreading model Anderson & May (1992) and the label indicates whether the node is infected or not. In the Airport dataset, the nodes represent airports with the population of the city as the label and the edges represents the flight routes between different cities. PubMed and Cora are citation network datasets where the nodes representing the scientific papers with the academic area as the label and the edges representing the existence of citations between papers. Table 1 shows the statistics for all datasets. The $\delta$-hyperbolicity represents the Gromovs hyperbolicity and is reported on these datasets by Chami et al. (2019). The lower the $\delta$-hyperbolicity value, the closer the graph to a tree i.e. more hyperbolic

Table 1: Datasets statistics

| Dataset | nodes | node features | nodes classes | edges | $\delta$-hyperbolicity |
|---------|-------|---------------|---------------|-------|------------------------|
| Disease | 1044  | 1000          | 2             | 1043  | 0                      |
| Airport | 3188  | 11            | 4             | 18631 | 1                      |
| PubMed  | 19717 | 500           | 3             | 88651 | 3.5                    |
| Cora    | 2708  | 1433          | 7             | 5429  | 11                     |

Table 2: Evaluation results and comparison with other methods. ROC AUC results are reported for Link Prediction (LP) tasks and F1 scores are reported for Node Classification (NC) tasks. The latent feature representation dimension is set to 16 for fair comparison.

| | Dataset | Disease | | Airport | | PubMed | | Cora | |
|---|---------|---------|---------|---------|---------|---------|---------|---------|---------|
| | **Method** | LP | NC | LP | NC | LP | NC | LP | NC |
| Euclidean | GCN | 64.7±0.5 | 69.7±0.4 | 89.3±0.4 | 81.4±0.6 | 91.1±0.5 | 78.1±0.2 | 90.4±0.2 | 81.3±0.3 |
| | GAT | 69.8±0.3 | 70.4±0.4 | 90.5±0.3 | 81.5±0.3 | 91.2±0.1 | 79.0±0.3 | 93.7±0.1 | 83.0±0.7 |
| | SAGE | 65.9±0.3 | 69.1±0.6 | 90.4±0.5 | 82.1±0.5 | 86.2±1.0 | 77.4±2.2 | 85.5±0.6 | 77.9±2.4 |
| | SGC | 65.1±0.2 | 69.5±0.2 | 89.8±0.3 | 80.6±0.1 | 94.1±0.0 | 78.9±0.0 | 91.5±0.1 | 81.0±0.1 |
| Hyperbolic | HGCN | 91.2±0.6 | 82.8±0.8 | 96.4±0.1 | 90.6±0.2 | 96.1±0.2 | 78.4±0.4 | 93.1±0.4 | 81.3±0.6 |
| | HAT | 91.8±0.5 | 83.6±0.9 | - | - | 96.0±0.3 | 78.6±0.5 | 93.0±0.3 | 83.1±0.6 |
| | LGCN | 96.6±0.6 | 84.4±0.8 | 96.0±0.6 | 90.9±1.7 | 96.8±0.1 | 78.6±0.7 | 93.6±0.3 | **83.3±0.7** |
| | HYPONET | 96.8±0.4 | **96.0±1.0** | 97.3±0.3 | 90.9±1.4 | 95.8±0.2 | 78.0±1.0 | 93.6±0.3 | 80.2±1.3 |
| | **SRBGCN** | **97.3±0.2** | 93.0±0.4 | **97.3±0.0** | **91.6±0.9** | **97.2±0.0** | **79.1±0.3** | **95.2±0.0** | 82.9±0.2 |

where a tree structure has a $\delta$-hyperbolicity value of zero as the case for the Disease dataset. The higher the $\delta$-hyperbolicity value, the closer the graph to a complete graph.

## 5.2 EVALUATION METRICS

F1 score is used as the evaluation metric for node classification tasks and the Area Under Curve (AUC) as the evaluation metric for link prediction tasks. 10 independent runs are performed and the mean and the standard deviation for each experiment are reported with the same data splits as in Errica et al. (2020) that were used in previous works. The code will be publicly available to be used by other researchers in the community to encourage further developments in this field. The code was developed using PyTorch Paszke et al. (2019) and experiments were run on Tesla P100 and Tesla T4 GPUs.

## 5.3 EVALUATION RESULTS AND COMPARISONS WITH OTHER GRAPH METHODS

2 shows the performance of different methods including Euclidean and hyperbolic ones. The Euclidean methods include GCN Kipf & Welling (2017), GAT Velickovic et al. (2018), SAGE Hamilton et al. (2017) and SGC Wu et al. (2019). HGCN Chami et al. (2019), HAT Zhang et al. (2021a), LGCN Zhang et al. (2021b) and HYPONET Chen et al. (2022) are the hyperbolic methods used in the comparison. The hyperbolic methods outperform the Euclidean ones specially for the tree-like Disease dataset which has a zero $\delta$-hyperbolicity. This proves the effectiveness of using the hyperbolic space to model graph data specially graphs with small Gromovs hyperbolicity. As shown in the table, SRBGCN outperforms all other methods for most of the benchmarks. Even for the other benchmarks that SRBGCN do not get the best performance using a latent representation of 16, our method can still get comparable results that are much better than other methods in an efficient way. This is a clear evidence of the advantage of learning the graph features directly in the hyperbolic space. It is worth mentioning that HAT and LGCN methods use techniques such as attention modules to improve the performance. Our method is simple yet very effective. For the disease dataset that has a tree structure with depth of 4, our method achieved a very good performance using latent feature representation dimension of 4 or 8 compared to other methods which shows the effectiveness of using the intrinsic transformations on specially tree-like datasets. This in turn can help in build-

Table 3: Comparison between different methods using different dimensions (dim) on the Disease and Cora datasets for the node classification task.

| Dataset | dim | GAT | HGCN | HAT | LGCN | HYPONET | **SRBGCN** |
|---------|-----|-----|------|-----|------|---------|------------|
| Disease | 4 | 49.4±6.3 | 73.2±6.5 | - | 87.4±3.1 | 91.0±3.8 | **93.1±0.3** |
|         | 8 | 76.7±0.7 | 81.5±1.3 | 82.3±1.2 | 82.9±1.2 | 92.9±1.0 | **93.3±0.4** |
| Cora | 64 | 83.1±0.6 | 82.1±0.7 | 83.1±0.5 | 83.5±0.5 | 81.5±0.9 | **83.8±0.3** |

Table 4: **Ablation study** on different datasets to show the effect of using only the spatial rotation (SR) transformation operation and both the spatial rotation and the boost (SR and B) operations. ROC AUC results are reported for Link Prediction (LP) tasks and F1 scores are reported for Node Classification (NC) tasks.

| Dataset | Disease | | Airport | | PubMed | | Cora | |
|---------|---------|----|---------|----|--------|----|------|----|
| **Transformation** | LP | NC | LP | NC | LP | NC | LP | NC |
| SR only $\mathbf{Y}_h^l = \mathbf{X}_h^l \mathbf{P}^l$ | 97.2±0.3 | 92.1±0.6 | **97.3±0.0** | 89.7±1.4 | **97.2±0.0** | 78.8±0.4 | **95.2±0.0** | 81.4±0.4 |
| SR and B $\mathbf{Y}_h^l = \mathbf{X}_h^l \mathbf{P}^l \mathbf{L}^l$ | **97.3±0.2** | **93.3±0.4** | 96.8±0.0 | **91.6±0.9** | **97.2±0.0** | **79.1±0.3** | 94.3±0.0 | **83.8±0.3** |

ing more compact models for such datasets. For larger and more complicated datasets with higher $\delta$-hyperbolicity such as Cora dataset, our method achieved better performance than other methods using higher dimensional latent space to embed the features for such more complicated datasets. Table 3 shows this comparison between the different methods.

## 5.4 ABLATION STUDY

We present an ablation study to show the effectiveness of using both the boost operation and the spatial rotation operation which are the decomposition of the Lorentz transformation matrix. Table 4 shows the performance comparison between using the transformation $\mathbf{Y}_h^l = \mathbf{X}_h^l \mathbf{P}^l$ and using the full transformation $\mathbf{Y}_h^l = \mathbf{X}_h^l \mathbf{P}^l \mathbf{L}^l$. Using both the boost and the spatial rotation operations i.e. the full Lorentz transformation usually gives better performance and increases the model expressiveness of the hyperbolic network specially for the node classification tasks with deeper networks.

## 5.5 DISTORTION

The average distortion can be computed as: $\frac{1}{n^2} \sum_{i,j}^n \left( \left( \frac{ned_{i,j}}{ngd_{i,j}} \right)^2 - 1 \right)^2$ where $ned_{i,j}$ is the normalized embedding distance between nodes $i$ and $j$ and $ngd_{i,j}$ is the normalized graph distance between them. The distortion reflects how close the structure of the graph is preserved in the graph embeddings and the closer the value to zero, the less-distorted the features. Table 5 shows the distortion using GCN, HGCN and SRBGCN methods on the Disease and Airport datasets. The Euclidean GCN method introduces a lot of distortion compared to the hyperbolic ones. At the same time, our method has less distortion than the HGCN method which resorts to the tangent space to perform network operations. This shows the effectiveness of using the intrinsic Lorentz transformations to build

Table 5: Distortion values for Disease and Airport datasets.

| Dataset | GCN | HGCN | **SRBGCN** |
|---------|-----|------|------------|
| Disease | 67.92±54.91 | 1.04±0.55 | **0.35±0.03** |
| Airport | 175.02±216.90 | 1.39±0.64 | **0.27±0.00** |

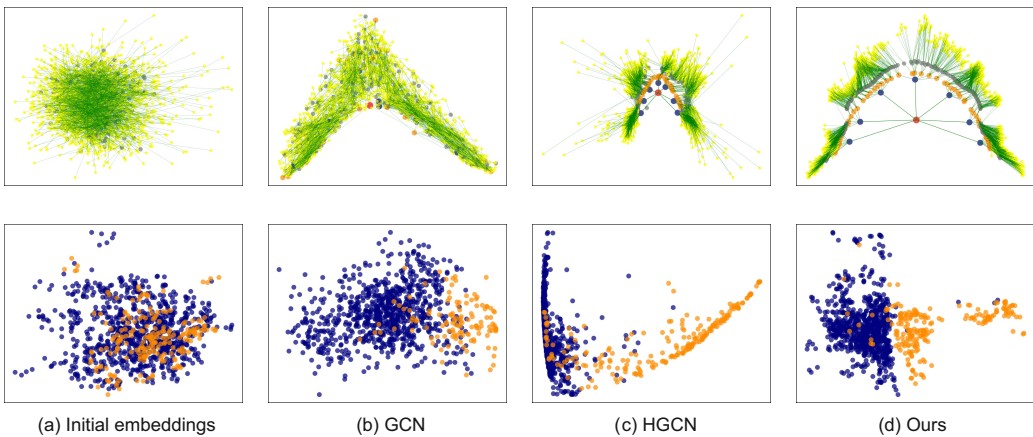

| (a) Initial embeddings | (b) GCN | (c) HGCN | (d) Ours |

Figure 4: The initial embeddings and the learnt embeddings using different methods on the whole Disease dataset for the link prediction (top row) and the node classification task (bottom row).

fully hyperbolic networks. Figure 4 shows the initial embeddings and the learnt embeddings in the last layer using these methods on the whole Disease dataset. For the link prediction task (top row), it is visually clear that the hyperbolic methods preserve the structure of the tree dataset better than the Euclidean method. Moreover, the hierarchies learnt from our method is much clearer than the one learnt by tangent-space methods. The visualization and distortion value show that our method generates higher quality features with less distortion. The ability to build better features with less distortion leads to higher performance which shows the superiority of our method. Similarly, for the node classification task (bottom row), our method separates the 2 classes of the Disease dataset more efficiently.

## 6 CONCLUSION

In this work, we presented a tangent-free full Lorentz transformation layer using the polar decomposition of the Lorentz transformation into both the boost operation and the spatial rotation operation. Our results show the effectiveness of using hyperbolic space to model graph data and to learn less distorted useful features which can be used to build more expressive and compact models for graph learning tasks. We hope this work can be extended to cover and build a unified framework that include other geometries such as Euclidean and Spherical spaces in order to increase the flexibility of the model to embed more complicated structures.

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

## A   APPENDIX

### A.1   HYPERBOLIC GEOMETRY REVIEW

We give a quick review about hyperbolic geometry to make the paper self-contained.

### A.1.1   TOPOLOGICAL SPACE AND TOPOLOGICAL HOMEOMORPHISM

A topological space is a geometrical space which has the notion of closeness. The closeness can, but not necessarily, be measured by the notion of distance to determine if points are close to each other. A homeomorphism is a continuous one-to-one mapping function or a bicontinuous function between topological spaces that has a continuous inverse function.

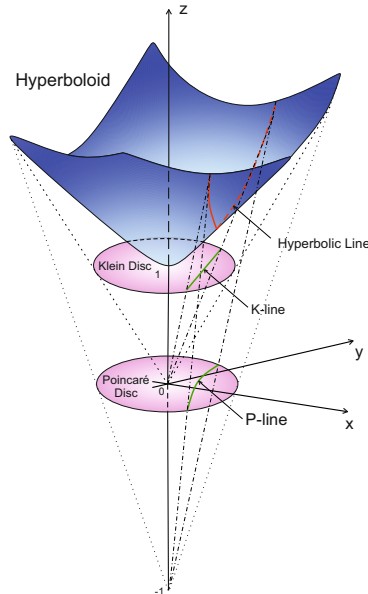

Figure 5: Projection of a hyperbolic geodesic from $\mathbb{H}^{2,K}$ onto the Klein disk and the Poincaré disk.

### A.1.2 MANIFOLD AND TANGENT SPACE

A $d$-dimensional Manifold $\mathcal{M}^d$ (which can be embedded in $\mathbb{R}^{d+1}$) is a topological space which can be locally approximated by a $d$-dimensional Euclidean space $\mathbb{R}^d$. For any point $x \in \mathcal{M}^d$, there is a homeomorphism between the neighbourhood of $x$ and the Euclidean space $\mathbb{R}^d$. Lines and circles are examples of one-dimensional manifolds. Planes and spheres are examples of two-dimensional manifolds which are called surfaces. The notion of manifold is a generalization of surfaces in any dimension $d$. The tangent space $\mathcal{T}_x\mathcal{M}^d$ at point $x \in \mathcal{M}^d$ is a $d$-dimensional hyperplane which is embedded in $\mathbb{R}^{d+1}$ that locally approximates the manifold $\mathcal{M}^d$ around the point $x$.

### A.1.3 RIEMANNIAN METRIC AND RIEMANNIAN MANIFOLD

A Riemannian metric $g$ is used to define geometric notions on the manifold such as distances, angles, areas or volumes. It is a collection of smoothly varying inner products on tangent spaces, $g_x : \mathcal{T}_x\mathcal{M}^d \times \mathcal{T}_x\mathcal{M}^d \to \mathbb{R}$. A Riemannian manifold can then be defined as $(\mathcal{M}^d, g)$.

### A.1.4 CURVATURE AND GEODESICS

A curvature measures how much a curve deviates from being a straight line. Euclidean spaces have zero curvature whereas non-Euclidean spaces have non-zero curvature For example, spheres have constant positive curvatures whereas hyperbolic spaces have constant negative curvatures. Geodesics are the generalizations of shortest paths in graphs or lines in Euclidean geometry to non-Euclidean geometry. These are the curves that give the shortest paths between pairs of points.

### A.1.5 HYPERBOLIC SPACE

A hyperbolic space is a Riemannian manifold with a constant negative curvature. Many models have been proposed to model a hyperbolic space such as the Lorentz model (also called the hyperboloid model), the Poincaré model and the Klein model. The Lorentz model is the upper sheet of a two-sheeted hyperboloid. The Poincaré model and the Klein model are the projections of the Lorentz model onto the hyperplanes $x_0 = 0$ and $x_0 = 1$, respectively. There are bijection functions to map between different hyperbolic models as they are all isomorphic. Figure 5 shows the three different models which model the hyperbolic space.

Table 6: Hyperparameters used for network training.

| Dataset | Disease | | Airport | | PubMed | | Cora | |
|---|---|---|---|---|---|---|---|---|
| **Parameter** | LP | NC | LP | NC | LP | NC | LP | NC |
| Learning rate | 0.001 | 0.005 | 0.5 | 0.2 | 0.05 | 0.04 | 0.001 | 0.001 |
| Number of layers | 2 | 6 | 1 | 2 | 1 | 5 | 1 | 3 |
| Weight decay | 0.0 | 0.0 | 1e-05 | 0.0 | 0.0 | 0.01 | 1e-04 | 0.01 |
| Dropout | 0.0 | 0.0 | 0.0 | 0.6 | 0.5 | 0.8 | 0.7 | 0.9 |
| Margin | 2 | 2 | 0.1 | 2 | 0.1 | 1 | 0.1 | 2 |
| Normalize features | 0 | 1 | 1 | 0 | 1 | 1 | 1 | 1 |

Let $\langle .,. \rangle_{\mathcal{L}} : \mathbb{R}^{d+1} \times \mathbb{R}^{d+1} \to \mathbb{R}$ represents the Lorentz-Minkowski inner product where $\langle x, y \rangle_{\mathcal{L}} := \sum_{i=1}^{d} x_i y_i - x_0 y_0 = x^T g_{\mathcal{L}} y$ where $g_{\mathcal{L}} = diag(-1, 1, \ldots, 1)$ is a diagonal matrix that represents the Riemannian metric for the hyperbolic manifold. Let $\mathbb{H}^{d,K}$ be a $d$ dimensional hyperboloid model with a constant negative curvature $-1/K$ where $K > 0$. Then we have:

$$\mathbb{H}^{d,K} := \{x \in \mathbb{R}^{d+1} : \langle x, x \rangle_{\mathcal{L}} = -K, x_0 > 0\} \tag{11}$$

Note that $x_0 > 0$ to indicate the upper half of the hyperboloid manifold. In special relativity, $x_0$ is referred to as the time axis whereas the rest of axes are called the spatial axes.

### A.1.6 EXPONENTIAL AND LOGARITHMIC MAPS

The exponential and logarithmic maps are used to map between the hyperbolic space and the tangent space and represent a bijection between the tangent space at a point and the hyperboloid. The exponential map maps a point $v \in \mathcal{T}_x \mathbb{H}^{d,K}$ where $x \in \mathbb{H}^{d,K}$ to the hyperboloid $\mathbb{H}^{d,K}$ such that $v \neq 0$ and is defined as:

$$exp_x^K(v) = \cosh(\frac{\|v\|_{\mathcal{L}}}{\sqrt{K}})x + \sqrt{K}\sinh(\frac{\|v\|_{\mathcal{L}}}{\sqrt{K}})\frac{v}{\|v\|_{\mathcal{L}}} \tag{12}$$

where $\|v\|_{\mathcal{L}} = \sqrt{\langle v, v \rangle_{\mathcal{L}}}$ is the norm of $v$. The logarithmic map maps a point $y \in \mathbb{H}^{d,K}$ to the tangent space $\mathcal{T}_x \mathbb{H}^{d,K}$ centered at point $x \in \mathbb{H}^{d,K}$ such that $x \neq y$ and is defined as:

$$log_x^K(y) = d_{\mathcal{L}}^K(x, y)\frac{y + 1/K\langle x, y \rangle_{\mathcal{L}} x}{\|y + 1/K\langle x, y \rangle_{\mathcal{L}} x\|_{\mathcal{L}}} \tag{13}$$

where $d_{\mathcal{L}}^K(x, y)$ is the Minkowskian distance between two points $x$ and $y$ in $\mathbb{H}^{d,K}$ and is given by:

$$d_{\mathcal{L}}^K(x, y) = \sqrt{K}\operatorname{arcosh}(-\langle x, y \rangle_{\mathcal{L}}/K) \tag{14}$$

### A.2 HYPERPARAMETERS DETAILS

The hyperparameters used for network training are shown in Table 6.

