# OpenReview forum: "SRBGCN: Tangent space-Free Lorentz Transformations for Graph Feature Learning"
_ICLR.cc/2023/Conference — Submitted to ICLR 2023_

### Official Review · Reviewer_Ht7h · 2022-10-22

**Confidence:** 4
**Correctness:** 4
**Technical Novelty And Significance:** 2
**Empirical Novelty And Significance:** 2
**Recommendation:** 5

**Clarity, Quality, Novelty And Reproducibility:**

The description of the approach is clear in general. What is missing is a clear discussion about the differences with the baselines.

**Strength And Weaknesses:**

- The main strength of the paper is the simplicity of the approach which is well-explained. I believe the proposed approach could be a useful "trick" to know in the community that works on hyperbolic representations.

- The weaknesses of the paper are missing theoretical comparisons with refs [A,B]. In particular:

1) Can both the boost and spatial rotation operations also be expressed by existing approaches (when working in the tangent space)? If the answer is yes, does the proposed approach act as a regularizer by limiting the nature of possible operations?

2) What are the operations that the proposed approach can or cannot perform?

3) Related to the above questions, in the introduction, it is mentioned: "However, these works [A,B] performed the network operations in the tangent space of the manifold which is a Euclidean local approximation to the manifold at a point since the network operations such
as feature transformation and feature aggregation are not manifold-preserving. This makes it hard
to optimize deeper networks because of these back and forth mappings between the manifold and the tangent space, and limits the representation capabilities of the hyperbolic networks caused by distortion specially most of these works used the tangent space at the origin." The tangent space is supposed to preserve the metric of the manifold due to the Levi-Civita connection of the manifold. I then do not think that the issue is due to performing operations in the tangent space. What are the reasons of the worse performance of the baselines? Is it because of the approximations of the mappings from the manifold to the tangent space? Or because of operations that are performed or not performed in the learned linear mappings?

**Summary Of The Paper:**

Similar to refs [A,B], the paper proposes a new version of Hyperbolic Graph Convolutional Network (HGCN). The main difference with refs [A,B] is that, instead of learning a linear map in the tangent space of the manifold, the paper proposes to learn linear operators directly in the extrinsic geometry of the Lorentzian ambient space of the hyperboloid.
The learned linear operators use some boost and spatial rotation operations and directly map to the hyperboloid.

[A] Ines Chami, Zhitao Ying, Christopher Ré, and Jure Leskovec. Hyperbolic graph convolutional neural
networks. NeurIPS 2019.

[B] Qi Liu, Maximilian Nickel, and Douwe Kiela. Hyperbolic graph neural networks, NeurIPS 2019.

**Summary Of The Review:**

The proposed method is interesting and could be useful to the community. What is missing is a clear discussion about the differences with refs [A,B]. What is possible in refs [A,B] that is not possible in the proposed approach and vice versa? What are the reasons of the improved performance of the method? Either a theoretical or empirical justification would be useful to readers.

---

> ### Author Response · Authors · 2022-11-19
> **reply to reviewer's questions**
>
> Thank you for your comments which help us make our work better.
>
> 1- The answer is explored with more details in [1]. Basically, the operations on the tangent space give a subset of the full Lorentz transformation (specifically spatial rotation with no boost).
>
> 2- The proposed approach covers the full Lorentz transformations i.e., both spatial rotation and boost. The optimization is fully done on the manifold without the need to resort to a tangent space as described in the paper.
>
> 3- Both. The local approximation causes the learned features to be distorted especially if only the tangent space of the origin is used for the approximation and not the tangent space for each point (the distortion section in the experimental part shows this). Also, due to the less expressiveness of the model and not covering all Lorentz transformations. Another point can be the extra operations of mapping between the manifold and the tangent space which can introduce some numerical errors and make it more difficult to optimize deeper networks for some tasks.
>
>
>
> [1] Chen et al. 2022, Fully hyperbolic neural networks.

---

### Official Review · Reviewer_x9HY · 2022-10-24

**Confidence:** 4
**Correctness:** 1
**Technical Novelty And Significance:** 2
**Empirical Novelty And Significance:** Not applicable
**Recommendation:** 6

**Clarity, Quality, Novelty And Reproducibility:**

The paper is easy to read and follow. But the work is not easy to reproduce since the details of the experiments are not given, such as optimization method, number of layers, patience, and training strategies.

**Strength And Weaknesses:**

## Weakness

1. Unconvincing conclusions.

(1) Indeed, $W\otimes_H X$ cannot guarantee that the result still lives in the Lorentz model after transformation. However, the problem has been solved by LGCN; that is, we only need to transform the space-like dimension (with the origin as a reference point) or just use the rotation matrix. Any of these methods can map it back to the Lorentz model. Also, the first approach can contain the rotation and boost operations.

Therefore, according to the transformation, the author cannot derive the conclusion—the method based on tangent space is not manifold-preserve. Similarly, we can utilize the same strategy to achieve the aggregation in the tangent space and successfully map it back to the Lorentz model.

(2) The author claimed that "This makes it hard to optimize deeper networks because of these back-and-forth mappings between the manifold and the tangent space," which could be incorrect since the deep network does not necessarily require frequent maps. For example, in image tasks, the network is deep [1][2]. Besides, it is not hard for us to optimize the HGNN. Please run the HGCN code directly using more than two layers.

[1] Capturing implicit hierarchical structure in 3D biomedical images with self-supervised hyperbolic representations
[2] Hyperbolic Image Segmentation

2. Confusing experimental results.

- Could the author explain why the performance of the disease and Pubmed reported is different from the original work HGCN? Why not take their original results but present the results from LGCN?
- The usage of dimension in Table 3 seems to cherry-pick. Why is the disease compared in 8 and 4 dimensions while the Cora is compared in 64?


**Summary Of The Paper:**

This paper argues that methods based on tangent space have many problems, and therefore, avoid adopting this method. This paper proposes an approach to manifold-preserving Lorentz transformation.

**Summary Of The Review:**

This paper argues that methods based on tangent space have many problems, and therefore, avoid adopting this method. This paper proposes an approach to manifold-preserving Lorentz transformation. Although the paper is easy to follow, there is something that I am concerned about, including the claim, method, and experiments.

---

> ### Author Response · Authors · 2022-11-19
> **reply to reviewer's concerns**
>
> Thank you for your comments which help us make our work better.
>
> (1) 1- We polished the text more to make it clearer. We meant that they need the tangent space to perform the network operations and can not do them directly on the manifold as these operations are not manifold preserving.
>
> 2- We found that for some tasks, the deeper the network, the more the performance degrades. We polish the text here more to make it clearer.
>
> (2) 1- For the disease dataset, it is because the original paper used lower number of layers. In LGCN, they used more layers and the authors did parameters search as well and reported the best results. For the Pubmed NC dataset, note that the original HGCN paper used pretrained embeddings from the LP task and finetuned the embeddings on the NC task.
>
> 2- This is to show that for easier datasets especially which have a pure tree-like structure, we can take advantage of that and build more compact models with fewer dimensions. However, for more complicated datasets such as Cora (note the Gromovs-hyperbolocity value and dataset statistics), we can build bigger networks to increase the performance.
>
> (3) The details for hyperparameters are provided in the revised draft. Thanks for pointing this out.

---

> > ### Comment · Reviewer_x9HY · 2022-11-29
> > **Thanks for your comments**
> >
> > Thanks for the author's detailed response. And the modification is appreciated. I raise my score.

---

> > > ### Author Response · Authors · 2022-12-01
> > > **Thanks for your reply**
> > >
> > > Thank you for your time.

---

### Official Review · Reviewer_8Du2 · 2022-10-24

**Confidence:** 3
**Correctness:** 2
**Technical Novelty And Significance:** 2
**Empirical Novelty And Significance:** 3
**Recommendation:** 5

**Clarity, Quality, Novelty And Reproducibility:**

**Clarity**
---

I think the paper's clarity could be significantly improved. I have two main issues with clarity.

1) Due to the writing it is not clear what is known and what is new.
2) The lack of clarity makes it difficult to understand the method in details and some things seem incorrect.

For the first, section 4.2 has two proofs for which the statements are missing. However, I think both of these proofs are unnecessary as they statements they are proving are known results.

For the second, due to the missing details there are few things that seem incorrect. I detail them here.

**The first is in Figure 1**. The left figure in seems to show a hyperbolic rotation in which the point P (in blue) is rotated to the point P (in red). However, this is not a rotation in hyperbolic space, it is actually a boost. Specifically, the figure shows 1 dimensional hyperbolic space, in which the only rotations are given by $P = \begin{bmatrix} 1 & 0 \\\\ 0 & \pm 1 \end{bmatrix}$ (as given by Equation (5) in the paper as well). In higher dimensions, again, the distance (euclidean) from the origin to the point should not change in a Lorentz rotation. Maybe this is what is being represented in the figure, but due to lack of details this is not clear.

**Another detail that is missing is that the paper does not define what a Lorentz boost is.** This missing definition should be added in. Further the discussion on page 4 is not clear to me. Particularly the statement that the boost can be realized as rotations.

**The next issue is in Figure 3** and equation (9) here the paper claims to be different from Chen et al 2022. However, this seems to be the same as Equation (4) (ArXiv v3 of Chen et al 2022). Note the cosmetic difference between the two due to the fact that this paper considers the curvature to $(-1/K)$ whereas Chen et al denote it by $K$.

**The loss function is not clear**. For classification with more than two labels it is not clear what $\hat{d}$ is? Do we pick a class at random? Is it weighted average of the distance to other classes?

**An important missing detail is the training method** for their neural network. The paper does not mention any training details. The only place such a detail is present is when the paper says it enforces orthogonalization constraint. However, my question is how is this constraint enforced. Is the method projected gradient descent, Riemannian gradient descent, or some form of natural gradient that preserves the manifold.

**The title is misleading**. The initial embedding step uses the Tangent space to get an initial hyperbolic embedding. If the authors want to be truly free using of the Tangent space, I would recommend using one of the hyperbolic embedding techniques (Nickel and Kiela NeurIPS 2017, Nickel and Kiela ICML 2018, Sala, De Sa, Gu and Re ICML 2018, Sonthalia and Gilbert NeurIPS 2020)


**Novelty**
---

As mentioned it is unclear to me how new the ideas in the paper are. In particular, the decomposition is known already and without the training details, the new parameterization doesnt seem that novel. Also the differences to Chen et al. are not quite as clear.

**Reproducibility**
---

I think the paper could improve its reproducibility. In particular, by adding all of the hyper parameters and training details for the different networks.


**Questions**
---

I have a few questions for the authors.

1) It is not clear to me what section 5.5 is trying to say. Specifically, why should our learned network be an isometry. This is further unclear to me since the loss function is the trying to minimize the distance between an embedding and the centroid for its class. This would suggest to be some sort of Neural Collapse would be beneficial.

However, the authors claim that being a near isometry is what we want. If we want an isometry then why learn features at all? Couldn't we just use the graph?


**Strength And Weaknesses:**

**Strengths**
---

1) The method captures all Lorentz transformation. This is in contrast to Chen et al 2022 for which they show their parameterization captures all rotations and all boosts but not necessarily both together in one layer. However, this distinction may not be too important.

2) The method seems to outperform other methods. But the important are small and may vanish with changing hyperparameters.

**Weaknesses**
---

1) It is not clear to me how novel the method is. The decomposition into boosts and rotations is known (Moretti 2002). Further for the proofs in the texts, there are no statements, but statements being proved should be known (Moretii 2002, Chen et al 2022).

2) There quite a few missing details including training details, statements of theorems among other things.

**Summary Of The Paper:**

The paper looks at graph hyperbolic neural networks to learn maps from $\mathbb{L}^n \to \mathbb{L}^d$ (where $\mathbb{L}^d$ is $d-1$ dimensional Lorentz manifold. In particular, given a graph $G$ we learn features for each node using such a map and then use this for node classification and link prediction.

The novelty in the paper is in the parameterization of the maps from $\mathbb{L}^n \to \mathbb{L}^d$ (where $\mathbb{L}^d$. In particular, they use the parameterization given by Moretti 2002. They then show that such networks outperform other networks on a variety of standard datasets.

They also do an ablation study and show that their method learns near isometries.

**Summary Of The Review:**

In summary, I think the idea of doing hyperbolic graph neural networks without using the tangent space is important since it increases the flexibility, and stability of the method as well as reduces computational costs.

However, it is not clear to be (due to the presentation) the novelty of the paper, the correctness of some of the claims, and the reproducibility of the results. I think these issues are fixable, but very much do need to be fixed.

---

> ### Author Response · Authors · 2022-11-19
> **Reply to reviewer's concerns and questions**
>
> Thank you for your comments which help us make our work better.
>
> 1- We think you mean the right figure and indeed it is a boost operation not spatial rotation. In this figure, we show that the boost operation can be realized by a hyperbolic rotation also known as squeeze mapping which is shown in 1 dimensional in the figure for illustration purposes. The regular/ Euclidean rotation can on the other hand be used to realize the spatial rotation operation in the hyperbolic space (left figure).
>
> 2- The boost operation can be realized by a cascade of basic hyperbolic rotations or by the reparameterization we gave which has a counterpart in the Euclidean case (rotation using a rotation axis and a rotation parameter.) The latter has the advantage of being more compact. Note that we cannot use the discontinuous space Euclidean rotations in our model. However, the hyperbolic rotations do not suffer from this problem and that is why we were able to successfully use it in our work.
>
> 3- They are the same and you are right about that. We may have based this observation on earlier codes released before. Thank you for this valuable observation and we updated this figure.
>
> 4- It is the distance to any non-correct class. Then to compute the total loss, we take the mean over all nodes and classes.
>
> 5- Enforcing the orthogonalization has been done before in orthogonal networks to basically learn orthogonal kernels. This is done by basically optimizing on the Stiefel manifold using a caylay retractor for example.
>
> 6- You have a point here. However, this applies only to the initial embeddings. So, the method itself is tangent-free and apparently if the initial imbedding is hyperbolic, it will be completely tangent-free. We can think of it as a preprocessing step.
>
> 7- The details for hyperparameters are provided in the revised draft. Thanks for pointing this out. For the difference with Chen et al. 22, their method is great and it can include both boost and spatial rotation operations. However, if we think about this method, we find it as a normalization method (basically linear transformation in the ambient Euclidean space, then followed by normalization to make sure the points are on the manifold.) Our method directly transforms the points on the manifold without normalization. Moreover, the reparameterization technique in our work is easily more physically interpretable and also can be easily extended to spatio-temporal or dynamic graphs. The experimental part suggests that indeed this method can get comparable or better results compared to other methods.
>
> 8- Thank you for your question. Please note that this experiment is done on the link prediction task only which makes sense here. The goal here is not to learn an isometry and again we need to predict links so feature learning is needed. In this part, we show that the learnt features preserve the hierarchies in the original graph.

---

> > ### Comment · Reviewer_8Du2 · 2022-11-29
> > **Thanks**
> >
> > 1) I am still a little confused about the right figure in Figure 1. As you have said you agree with me that it is a boost. But I think it currently labelled as a hyperbolic rotation.
> >
> > 2) This statement is not clear to me "The boost operation can be realized by a cascade of basic hyperbolic rotations" are you saying if $L$ is a Lorentz boost, then there is an infinite sequence of Lorentz rotations $R_i$ such that $R_1 \circ R_2 \circ \ldots \circ R_n \to L$ as $n \to \infty$. If that this is the case could you provide a reference for this?
> >
> > 3) Thanks
> >
> > 4) Thanks for the clarification
> >
> > 5) Thanks for the clarification
> >
> > 6) Thanks
> >
> > 7) Thank you for adding the hyper parameters. You are right that Chen et al 22 can be viewed as a form of normalization and that maintaining the orthogonality, is a method of parameterization. However, as the paper is written this seems to be focused on discussing what happens in functions space (in which they are the same). However, the experiments would suggest that the difference in the parameterization has an affect. I would recommend that the authors focus on this. This would help highlight the differences.
> >
> > 8) Thanks.
> >
> > While the authors, have addressed most of my concerns, (concern 1 and 2 should also be easily addressable). However, it is concern 7 that I think requires some work. I think this is distinction between the current work and the Chen et all 22 work needs to be better developed and highlighted. As such I will increase my score to a 5, but am still leaning towards reject.

---

> > > ### Author Response · Authors · 2022-12-01
> > > **reply and thanks**
> > >
> > > Thank you for your reply and comments which help us make our work better.
> > >
> > > 1-Yes, hyperbolic rotation means moving a point along a hyperbola. Boost means moving a point on the hyperboloid along time axis without rotating the spatial axes. So, boost can be realized by a hyperbolic rotation. We used a parameterization of a hyperbolic rotation axis and a hyperbolic rotation parameter. So, this L matrix (equation 6) moves the point along a hyperbola in a plane parallel to the plane formed by the hyperbolic rotation axis and the time axis by the hyperbolic rotation parameter. Similarly, the spatial rotation operation can be implemented by regular rotations.
> > >
> > > 2-A finite sequence of hyperbolic rotations (d rotations for a d-dimensional hyperboloid). That is, a hyperbolic rotation on a hyperbola in a plane parallel to the plane formed by a basis spatial rotation axis and the time axis. We have d basis spatial rotation axes and hence d total possible basic hyperbolic rotations. We can try to polish the text more here regarding the hyperbolic rotations to make it clearer.
> > >
> > > 3-Thank you for raising this point. Another difference which your comment has raised is that in our method, we use 2 steps with a decomposition that is manifold-preserving. For chen’s method, it is 2 steps with the second one as a normalization step. In our case, we can use only one step which is manifold-preserving and still get a good performance. However, it is essential for chen’s methods to keep the points on the manifold otherwise, the performance degrades. This can be shown using an experiment but chen’s method would then not be a hyperbolic manifold-preserving method then. There are other differences between the two methods as pointed out before. We are happy for this discussion and any further comments or suggestions would be appreciated. Thanks for your time.

---

### Official Review · Reviewer_ehJR · 2022-10-25

**Confidence:** 5
**Correctness:** 3
**Technical Novelty And Significance:** 1
**Empirical Novelty And Significance:** 1
**Recommendation:** 3

**Clarity, Quality, Novelty And Reproducibility:**

The paper/methodology is generally easy to follow and understand. Lack of originality, reproducibility not clear.

**Strength And Weaknesses:**

There are recently many efforts on removing the tangent space of hyperbolic neural networks. This paper used the Lorentz boost and rotation to operate directly on the Lorentz manifold.

Strength. The presentation and logic of the paper are easy to follow.
The ablation study is interesting and shows the learnt structure information from the model.

Weakness.
1. Lack of novelty and contribution. The Lorentz boost and rotation is already well-characterized and studied in [1], particularly the proposed fully hyperbolic linear layer in [1] contains all Lorentz rotation and boost matrices. In fact, the graph convolution operation is essentially a hyperbolic linear layer. One can just take the linear layer from [1] and make a similar tangent space free Lorentz based hyperbolic GCN, which is pretty much done in the paper.

2. Limited experiments and lack of reproducibility. Experiments are performed on 4 standard small scale datasets, which are 4 highly overfitted datasets. For example, the disease and airport, they are so small datasets. From my experience, I can always get a very high accuracy, but with a large variance during multiple runs. No code or supplementary is provided for the model. Plus I can get higher accuracies with some baselines, for example, using HGCN on pubmed node classification, $80.2\pm 0.3$ with 10 runs. It's really hard to derive meaningful conclusion from the experiments.

------ 2.1 where is the training hyper-parameter? how many layers of the model, any regularization, hidden dimension? How many epochs? I am aware that many hyperbolic NNs report results of running thousand epochs with early stopping, while standard Euclidean GCNs, SGC report accuracy of 100 or 200 epochs. Is the comparison fair?

3. Some claims are vague or inaccurate. For example,

"most of the existing hyperbolic networks build the network operations on the tangent space of the manifold, which is a Euclidean local approximation. This distorts the learnt features, limits the representation capacity of the network"

I don't think this holds true, generally the exp/log map is local approximation of the manifold. However, in the hyperbolic space, they are bijection between the tangent space and the hyperbolic space. Usage of exp map at origin is not a problem with respect to representation capacity, as one can use parallel transport to move it along the manifold, which is already introduced and used in many prior works.

------ 3.1 It claimed that "the full Lorentz transformation ... increase the model expressiveness ... with deeper networks." I failed to find evidence in the paper supporting this claim, in particular, personally I haven't seen any successful deep hyperbolic GCN style structure to work well even in existing/prior work.

4. The word and text needs more polish. For example,

"This makes it hard to optimize deeper networks because of these back and forth mappings between the manifold and the tangent space, and limits the representation capabilities of the hyperbolic networks caused by distortion specially most of these works used the tangent space at the origin. "

[1] Chen et al. 2022, Fully hyperbolic neural networks.

**Summary Of The Paper:**

This paper proposes a version of hyperbolic GCN based Lorentz model without resorting to the tangent space. Some of the layer operations exist in prior work. Experiments are conducted on standard small-scale citation networks and tree datasets.

**Summary Of The Review:**

It's a straightforward usage of the fully hyperbolic NNs paper in GCN applications. Many pieces in the paper are missing and lack of enough novelty, see above.

---

> ### Author Response · Authors · 2022-11-19
> **Reply to the reviewer's concerns**
>
> Thank you for your comments which help us make our work better.
>
> 1- The work in [1] is a great work and indeed the linear layer they used includes the boost and rotation operations as they proved. However, if we think about this method, we find it as a normalization method (basically linear transformation in the ambient Euclidean space, then followed by normalization to make sure the points are on the manifold.) Our method directly transforms the points on the manifold without normalization. Moreover, the reparameterization technique in our work is easily more physically interpretable and also can be easily extended to spatio-temporal or dynamic graphs. The experimental part suggests that indeed this method can get comparable or better results compared to other methods.
>
> 2- We attach the code with experimental details and hyperparameters as supplementary for reproducibility. For the datasets used in the experiments, most of other works used these datasets to test the performance of their methods. For HGCN on pubmed NC, this higher accuracy can be obtained using the embeddings learnt on the LP task and then finetuning on the NC task. We reported the results from [2] and in this work they have done parameter search for all methods they used in the comparison.
>
> 2.1- It is true that the number of epochs in subsequent non-Euclidean work is far more the Euclidean counterpart but if we do the same for the Euclidean ones and run them for more epoches, the accuracy will still be limited and will not outperform the other ones. We can get some improvements but will not outperform the other methods. The details for hyperparameters are provided in the revised draft.
>
> 3- The visualization in our work illustrates the embeddings learnt by the tangent-space HGCN method and we can see the quality of the learnt embeddings is not as good as the fully hyperbolic one. This also can be attributed to the fact that the tangent space operations are subset of the full Lorentz transformations [1]. Another point can be as a result of those extra operations (exp/ log) which can affect the backpropagation step with additional numerical errors. Building deeper networks using tangent-space operations generally degrades the performance which again can be attributed to those operations and in turn, limits the representation capacity of the network.
>
> 3.1- As we can get rid of those extra tangent-space operations, we can build deeper networks with more layers. For example, in this work the Disease NC model has 6 layers.
>
> 4- Thank you for pointing this out, we polish the text more to make it better for understanding.
>
>
> [1] Chen et al. 2022, Fully hyperbolic neural networks.
> [2] Zhang et al. 2021, Lorentzian graph convolutional networks

---

### Decision · Program_Chairs · 2023-01-20

**Decision:**

Reject

**Justification For Why Not Higher Score:**

limited novelty

**Justification For Why Not Lower Score:**

N/A

**Metareview: Summary, Strengths And Weaknesses:**

The paper proposes a "fully hyperbolic" GNN architecture (i.e., without resorting to the tangent space as done typically in manifold optimization). This seems to be the main novelty of the paper, which is incremental in light of prior work cited by the reviewers (Chen et al 2022 and Moretti 2002). Experiments are also rather limited and not particularly convincing. We recommend rejection.